# A Modified Preassigned Finite-Time Control Scheme for Spacecraft Large-Angle Attitude Maneuvering and Tracking

**DOI:** 10.3390/s25030986

**Published:** 2025-02-06

**Authors:** Xudong Ma, Yuan Liu, Yi Cheng, Kun Zhao

**Affiliations:** 1School of Aeronautics and Astronautics, Sun Yat-sen University, Shenzhen 518107, China; maxd8@mail2.sysu.edu.cn; 2Shanghai Institute of Satellite Engineering, Shanghai 201109, China; frank.cheng@foxmail.com (Y.C.); zk1988918@hotmail.com (K.Z.)

**Keywords:** large-angle attitude maneuver, attitude tracking, angular velocity constraints, control torque constraints

## Abstract

This paper addresses the problem of large-angle attitude maneuvering and tracking control for rigid spacecraft, considering angular velocity and torque constraints, actuator faults, and external disturbances. First, a sliding-mode-like vector is constructed to guarantee the satisfaction of the angular velocity constraints. A modified preassigned finite-time function, which can adaptively adjust the boundaries, is then proposed to constrain the sliding-mode-like vector. The controller is designed to stabilize the closed-loop system using a barrier Lyapunov function. Additionally, actuator saturation is compensated adaptively, and the system’s lumped disturbance is estimated using a fixed-time disturbance observer. Finally, the practically preassigned finite-time stability of the closed-loop system is demonstrated. In practical applications, the proposed controller can guarantee transient and steady-state performance, prevent excessive angular velocity, and ensure compliance with the physical limitations of the actuators. Simulation results are provided to demonstrate the effectiveness of the proposed controller.

## 1. Introduction

Due to their critical role in space applications, such as in-orbit servicing [1], debris removal [2,3], and space habitat designs [4,5,6], attitude maneuvering and tracking have received significant attention in recent years. Spacecraft are required to execute rapid, large-angle maneuvering and precise pointing to meet the demands of various missions [7,8,9]. Designing a controller that ensures fast convergence, high tracking precision, and robust disturbance rejection becomes an important issue due to the nonlinearity of the spacecraft dynamics, parameter uncertainties, and external disturbances. Various control techniques, including adaptive control [10,11,12], classical sliding-mode control [13,14,15], and prescribed performance control (PPC) [16,17,18,19], have been proposed to improve the performance of closed-loop systems. Although these methods achieve excellent performance, the equilibrium is asymptotically stable, meaning that the convergence time tends to infinity, which limits their applicability in real-time engineering systems.

Finite-time control has gathered a great deal of attention in recent years. In contrast to asymptotic controllers, finite-time controllers ensure the stability of closed-loop systems within finite time intervals and provide better performance in terms of tracking accuracy and disturbance rejection. Consequently, several approaches have been proposed for spacecraft tracking and stabilization, including the homogeneous-based method [20], finite-time stability-based method [21], and terminal sliding-mode control method [22]. Based on the homogenous method, a finite-time controller for attitude stabilization was designed, considering actuator saturation [23]. Additionally, Ref. [24] proposed the practically finite-time stability (FTS) theory to address system disturbances and uncertainties. A finite-time controller was designed for attitude tracking using the power integrator technique [25], integrated with FTS theory. To address parameter uncertainties and external disturbances, Wu et al. [26] introduced a finite-time controller based on the backstepping technique, combined with an extended state observer. Due to its superior disturbance rejection properties, the terminal sliding-mode control method has been utilized in the design of efficient finite-time controllers. An adaptive fast terminal sliding-mode controller has also been proposed to address the attitude tracking control problem in the presence of actuator faults, ensuring finite-time convergence [27]. In combination with the adaptive law for lumped uncertainty, a nonsingular terminal sliding controller has been proposed for attitude tracking of rigid spacecraft. The finite-time stability of the closed-loop system has been demonstrated using the Lyapunov stability theory [28]. While these finite-time controllers provide better convergence performance, the settling time remains dependent on the initial conditions and controller parameters. In such applications, it is crucial to ensure that the system remains stable within a predefined time, independent of initial conditions. Furthermore, while previous studies have primarily focused on steady-state tracking errors, transient performance also plays a critical role in attitude control. The performance constraints have therefore become an important research topic.

Funnel control, initially proposed by Ilchman et al. [29], is a method designed to ensure prescribed transient performance and applied to linear MIMO systems [30]. In Ref. [31], a new funnel parameter was proposed to generate high gain when the state approaches the funnel boundary, ensuring the transient performance. Funnel control was developed to overcome the difficulties of identification and estimation of traditional high-gain adaptive control. Nevertheless, funnel control is typically applied for a specific class of linear or nonlinear systems with relative degrees of ‘one’ or ‘two’. This limits its broader applicability in more general systems.

Inspired by funnel control methods, the prescribed performance control (PPC) methodology, initially introduced by Bechlioulis [32], is widely used to constrain attitude tracking errors within a predefined envelope. This approach has demonstrated significant effectiveness in spacecraft attitude control. Hu et al. [16] extended the PPC to address attitude tracking problems in the presence of actuator faults and input saturation. Similarly, Ref. [33] focused on flexible spacecraft attitude tracking and proposed a mode-free PPC law that guarantees performance in both transient and steady states. However, one limitation of this approach is that the time required to reach the steady-state boundary cannot be preassigned. To overcome this limitation, the concepts of Prescribed Finite-Time Stability (PFTS) and preassigned finite-time function (PFTF) were proposed [34]. Unlike conventional finite-time controllers, the prescribed finite-time controller allows the settling time to be preset and remains independent of both the initial conditions and design parameters, thus enhancing its flexibility and applicability. Furthermore, the preassigned transient and steady-state performance can be effectively achieved. The sufficient condition of FTS in Ref. [24] need not be satisfied, which simplifies the controller design process. Based on the PFTF [35], a preassigned finite-time controller was proposed for attitude tracking in the presence of parameter uncertainties and external disturbances, with the practically preassigned finite-time stability of the closed-loop system demonstrated. Gao et al. [36] designed a controller with a preset time that ensures both transient and steady-state performance in the presence of actuator saturation and faults. In Ref. [37], an adaptive prescribed finite-time controller was proposed for a class of strict-feedback systems. As a breakthrough in finite-time control theory, the preassigned finite-time control scheme offers several advantages. Fractional powers are unnecessary for controller design and stability analysis, simplifying both the design process and the selection of controller parameters. Moreover, the convergence time of the closed-loop system is predetermined and remains independent of the initial conditions and other control parameters.

A barrier Lyapunov function (BLF) is also one of the popular constraint control strategies [38,39,40], originally proposed by Keng et al. [38]. Unlike PPC, the BLF uses special types of functions, such as log functions, which tend to infinity as the constrained variables approach the boundaries. The objective of constraining the variables is achieved by ensuring that the BLF remains bounded. In Ref. [7], the BLF, combined with backstepping methods, was applied to the design of an attitude tracking controller for a noncooperative fly-around mission, ensuring that state constraints are satisfied. In Ref. [41], an adaptive dynamics surface control method incorporating the BLF was proposed to address the state constraints problem. However, a limitation of the BLF method is that when the Lyapunov function or constraint boundaries are adjusted, the controller must be redesigned, and the stability conditions of the closed-loop system need to be reestablished.

Most of the existing literature overlooks the constraints on angular velocity and torque in preassigned finite-time control strategies for spacecraft tracking and large-angle maneuvering. In Ref. [35], these limitations were not addressed, while in Refs. [36,42], only torque input limitations were considered. In practical engineering applications, angular velocity and torque inputs are constrained by the physical limitations of the reaction wheel and rate gyroscope. Excessive angular velocity, in particular, could induce flexible vibrations, and actuator saturation may lead to closed-loop system instability. The preassigned finite-time controllers proposed in the literature typically exhibit rapid convergence rates, causing the limits of angular velocity and torque to be exceeded. Furthermore, the tracking error could exceed the prescribed steady-state boundary in the presence of sudden external or internal disturbances, potentially resulting in control singularity and system instability.

Inspired by the literature, this paper proposes a novel attitude controller based on the BLF. The main contributions of this work can be summarized as follows:(1)In contrast to previously developed preassigned finite-time control strategies [35,36], both angular velocity and torque constraints are explicitly addressed in this paper. The angular velocity constraint is incorporated by constructing a sliding-mode-like vector, which is subsequently constrained using the performance function. Additionally, an adaptive term is introduced to compensate for actuator saturation.(2)A modified preassigned finite-time function, incorporating an adaptive term, is introduced to address the control singularity caused by sudden external or internal disturbances during the steady state. The issue is not addressed in most of the current literature on preassigned finite-time control methods (e.g., [23,24,28]).

The remainer of the paper is organized as follows. Section 2 presents the spacecraft dynamics model and control objective, with the proposed modified preassigned finite-time control scheme. Section 3 presents simulation results to assess the effectiveness of the proposed method, followed by a discussion of the results. Section 4 concludes the paper.

## 2. Methods

### 2.1. Spacecraft Model and Problem Formulation

#### 2.1.1. Attitude Dynamics and Kinematics Model

The attitude kinematics and dynamics of rigid spacecraft can be described as follows [36]:(1)$$\begin{array}{c} {\dot{q}}_{v}=\frac{1}{2}\left(q_{0}I_{3}+q_{v}^{\times }\right)w\\ {\dot{q}}_{0}=-\frac{1}{2}q_{v}^{T}w \end{array}$$(2)$$J\dot{w}=-w^{\times }Jw+Eu_{a}+\bar{u}+d$$
where the unit quaternion $q={\left[q_{0},q_{v}^{T}\right]}^{T}\in R^{4}$ represents the attitude orientation in the body frame relative to the inertia frame, subject to the constraint $q_{v}^{T}q_{v}+q_{0}^{2}=1$. $w\in R^{3}$ represents the angular velocity of the spacecraft expressed in the body frame. The symmetric inertia matrix is represented by $J\in R^{3\times 3}$. The actuator’s efficiency matrix is given by $E=diag\{e_{1}\left(t\right),e_{2}\left(t\right),e_{3}\left(t\right)\}$, where $0\leq e_{i}\left(t\right)\leq 1$. The bounded disturbance is represented by $d\in R^{3}$, with $\left\|d\right\|\leq D$. $\bar{u}\in R^{3}$ represents unexpected deviation faults from the actuator.

#### 2.1.2. Actuator Saturation Model

The model of the actuator with saturation is defined as follows [43]:(3)$$u_{ai}=u_{imax}tanh(\frac{u_{ci}}{u_{imax}})$$

Let *i* = 1, 2, 3, where $u_{a}\in R^{3}$ is the commanded control input and $u_{c}\in R^{3}$ is an auxiliary intermediate signal to be designed. It is noted that $\left|u_{ai}\right|\leq u_{imax}$ for i=1, 2, 3 and the commanded control input $u_{a}$ satisfies the saturation limits.

#### 2.1.3. Attitude Error Dynamics and Kinematics

To complete the controller design, the attitude error dynamics and kinematics model are formulated. The desired attitude unit quaternion, $q_{d}={\left[q_{d0},q_{dv}^{T}\right]}^{T}\in R^{4}$, and $w_{d}\in R^{3}$ represent the desired angular velocity. The unit quaternion error is defined as $q_{e}={\left[q_{e0},q_{ev}^{T}\right]}^{T}\in R^{4}$, where $q_{e0}=q_{dv}^{T}q_{v}+q_{0}q_{d0}$ and $q_{ev}=q_{d0}q_{v}-q_{dv}^{\times }q_{v}-q_{0}q_{dv}$. The angular velocity error is defined as $w_{e}=w-R_{e}w_{d}$, where $R_{e}$ is given by $R_{e}=\left(q_{e0}^{2}-q_{ev}^{T}q_{ev}\right)I_{3}+2q_{ev}q_{ev}^{T}-2q_{ev}q_{ev}^{\times }$.

The attitude error kinematics model is then established as follows:(4)$$\begin{array}{c} {\dot{q}}_{ev}=\frac{1}{2}\left(q_{e0}I_{3}+q_{ev}^{\times }\right)w_{e}\\ {\dot{q}}_{e0}=-\frac{1}{2}q_{ev}^{T}w_{e} \end{array}$$

Based on Equations (2) and (3), the attitude error dynamics model is expressed as follows:(5)$${\dot{w}}_{e}=F+u+D_{f}$$
where $F=J^{-1}(-w^{\times }Jw+Jw_{e}^{\times }R_{e}w_{d}-JR_{e}{\dot{w}}_{d})$, $D_{f}=J^{-1}(\bar{u}+d+\left(E-I_{3}\right)u_{a})$, $u=J^{-1}{(u}_{c}+∆u)$, and ∆u = $u_{a}-u_{c}$.

#### 2.1.4. Control Objective

This paper aims to design a preassigned finite-time attitude controller that achieves the following objectives, even in the presence of external disturbances, actuator saturation, and faults:

(1)The attitude quaternion errors $q_{ev}(t)$ converge to a small residual set within a finite time, under the condition of angular velocity constraints.(2)The control singularity problem is avoided in the presence of sudden external or internal disturbances in the steady state.

Based on the above objectives, a block diagram of the closed-loop system presented in this study is shown in Figure 1.

In Figure 1, variable s is sliding-mode-like vector, proposed to limit the angular velocity and designed in Section 2.2.3. The modified preassigned finite-time function $\bar{\rho }$ is developed to constrain vector s in Section 2.2.1. The actuator faults and system external disturbance ${\hat{D}}_{f}$ are estimated using a disturbance observer in Section 2.2.2. The actuator compensation variable z is applied to design controller $u_{c}$ in Section 2.2.3. $u_{a}$ is the actual output of the actuator, which affects the spacecraft dynamics model.

### 2.2. Main Results and Analysis

In this section, the attitude controller is designed with the consideration of external disturbances, actuator saturation, and faults. First, the relevant definitions are provided as follows:

**Definition** **1**([34] (Practically Preassigned Finite-time Stability, PPFTS)). *Consider the following nonlinear system*:
(6)$$\dot{x}\left(t\right)=f(x\left(t\right),u(t))$$
*where*
$x\left(t\right)$
*and*
$u\left(t\right)$
*are the system state and control input, respectively. Practically preassigned finite-time stability is satisfied if there exist constants*
ζ>0
*and*
$T\in \left(0,\infty \right)$
*such that for any initial condition*
$x\left(0\right)=x_{0}$*, the solution satisfies the bound*
$\;\left\|x\left(t\right)\right\|\leq \zeta \;$*for all*
$t\geq t_{0}+T$. T
*denotes the preassigned finite time, and the solution of Equation (6) ensures practically preassigned finite-time stability.*

**Definition** **2**([34] (Preassigned finite-time function, PFTF)). *The smooth function*
ρ(t)
* is a preassigned finite-time function if* ρ(t)* satisfies the following properties: (1)*
$\rho \left(t\right)>0$*; (2)*
$\dot{\rho }\left(t\right)\leq 0$*; and (3)*
$\rho \left(t\right)=\rho_{Tf}$
*when*
$t\geq T_{f}$*, where*
$\rho_{Tf}$
*is the steady-state error, and*
$T_{f}$
*is the settling time*.

**Definition** **3**([24] (Practical finite-time stable, PFS)). *Consider the system (6). Suppose that there exist continuous function V(x) scalars*
λ>0*,*
0<α<1*, and*
0<η<∞
*such that* $\dot{V}\left(x\right)\leq -\lambda V^{\alpha }\left(x\right)+\eta$. *Then, the solution of system (6) is PFS and bounded in finite time* $T_{r}$
*as*
$\underset{t\to T_{r}}{\lim}x\in \left\{x\in R^{n}|V^{\alpha }\left(x\right)\leq \frac{\eta }{\theta }\right\}$, 0<θ<λ, $T_{r}\leq \frac{V^{1-\alpha }(x_{0})}{\left(\lambda -\theta )(1-\alpha \right)}$.

**Lemma** **1.***According to* [22]*, for all real numbers*
$x_{i}$
*, i = 1, 2,…, n, and*
0<l<1*, the following inequality holds*:(7)$${\left(\left|x_{1}\right|+\ldots +\left|x_{n}\right|\right)}^{l}<{\left|x_{1}\right|}^{l}+\ldots +{\left|x_{n}\right|}^{l}$$

**Lemma** **2.***According to* [44]*, if*
$\left|x\right|<f$*, the following inequality holds, where*
f>0.(8)$$ln\frac{f^{2}}{f^{2}-x^{2}}\leq \frac{x^{2}}{f^{2}-x^{2}}$$

**Lemma** **3.***According to* [45]*, for any*
$\mu_{0}>0$
*and for*
$\delta_{0}=0.2785$*, the following inequality is satisfied*:(9)$$0\leq \left|x\right|-xtanh(\frac{x}{\mu_{0}})\leq \delta_{0}\mu_{0}$$

**Assumption** **1**([46]). The external disturbance and its first derivative are bounded, i.e., $d,\dot{d}\in L_{\infty }$.

**Assumption** **2**([36]). The actuator fault $\bar{u}$ is bounded, i.e., $\bar{u}$ $\in L_{\infty }$.

#### 2.2.1. Modified Preassigned Finite-Time Function (MPFTF)

In this paper, the following PFTF [36] is introduced for the design of the robust controller:(10)$$\rho_{i}\left(t\right)=\left\{\begin{array}{c} \rho_{0i}e^{-\frac{\rho_{\lambda }t}{T_{i}-t}}+\rho_{Tfi}\;\;\;\;\;\;\;\;\;\;\;\;\;\;\;\;\;\;\;\;\;\;\;\;\;t\in [0,T_{i})\\ \rho_{Tfi}\;\;\;\;\;\;\;\;\;\;\;\;\;\;\;\;\;\;\;\;\;\;\;\;\;\;\;\;\;\;\;\;\;\;\;\;\;\;\;\;\;\;\;\;\;\;\;\;\;\;t\in [T_{i},\infty ) \end{array}\right.$$
where *i* = 1, 2, 3, and $T_{i}$ represents the settling time. The maximum overshoot is given by $\rho_{i}(0)=\rho_{i0}+\rho_{Tfi}$, where $\rho_{Tfi}$ represents the prescribed steady-state bound. According to [36], Equation (10) satisfies all properties defined in Definition 2.

When the error signal approaches the performance bounds, such as in the case of a sudden additional disturbance caused by the actuator fault, the control effort will increase. Due to actuator saturation, there may not be sufficient control input to maintain the tracking error within the predefined performance bounds, potentially leading to a control singularity problem. To address this issue, a novel modified preassigned finite-time performance function (MPFTF), based on Equation (10), is described as follows:(11)$${\bar{\rho }}_{i}\left(t\right)=\rho_{i}\left(t\right)+\varphi_{i}$$

Here, $\varphi \in R^{3}$ is an auxiliary term designed as follows:(12)$${\dot{\varphi }}_{i}=-\alpha_{1}\varphi_{i}+\alpha_{2}\psi_{i}$$
where $\alpha_{1}\;\mathrm{and}\;\alpha_{2}$ are positive constants. $\alpha_{1}$ denotes the convergence rate of the auxiliary signal and $\alpha_{2}$ regulates the sensitivity of $\varphi_{i}$ in response to the changes in the value of the error signal. $\psi_{i}$ is defined in Section 2.2.3. According to (12), $\varphi_{i}\left(t\right)=\varphi_{i}(0)e^{-\alpha_{1}t}+\int_{0}^{t}e^{-\alpha_{1}(t-\tau )}\alpha_{2}\psi_{i}d\tau$. If $0\leq \left|\psi_{i}\right|\leq 1$, it follows that $0\leq \varphi_{i}\left(t\right)\leq \varphi_{i}\left(0\right)e^{-\alpha_{1}t}+\frac{\alpha_{2}}{\alpha_{1}}(1-e^{-\alpha_{1}t})$, and we have $\varphi_{i}\left(t\right)\leq \frac{\alpha_{2}}{\alpha_{1}}$ if $\varphi_{i}\left(0\right)=0$.

#### 2.2.2. Disturbance Observer

To enhance robustness, a disturbance observer is introduced to provide more accurate estimations of the disturbance. Based on Equation (5), defining $x_{1}=w_{e}$ and $x_{2}=D_{f}$, and under Assumptions 1 and 2, the term ${\dot{x}}_{2}$ is bounded. The dynamics model in Equation (5) is then rewritten as follows:(13)$$\left\{\begin{array}{c} {\dot{x}}_{1}=x_{2}+F+u\\ {\dot{x}}_{2}={\dot{D}}_{f} \end{array}\right.$$

A fixed-time disturbance observer, as proposed in [47], is introduced and defined as follows:(14)$$\begin{array}{c} {\dot{z}}_{1i}=z_{2i}+k_{1i}\left({\left[\frac{x_{1i}-z_{1i}}{\varepsilon }\right]}^{\alpha_{i}}+{\left[\frac{x_{1i}-z_{1i}}{\varepsilon }\right]}^{\beta_{i}}\right)+F_{i}+u_{i}\\ {\dot{z}}_{2i}=\frac{k_{2i}}{\varepsilon }\left({\left[\frac{x_{1i}-z_{1i}}{\varepsilon }\right]}^{{2\alpha }_{i}-1}+{\left[\frac{x_{1i}-z_{1i}}{\varepsilon }\right]}^{{2\beta }_{i}-1}\right) \end{array}$$
where $z_{ji}$ represents the estimation of $x_{ji}$, $k_{ji}$ is a positive constant, and ε is a small adjustable parameter. Additionally, $\alpha_{i}\in (\frac{1}{2},1)$ and $\beta_{i}\in (1,\frac{3}{2})$, with j=1, 2 and i=1, 2, 3. According to [47], the disturbance estimation error is bounded within a fixed time, given by(15)$$T\leq \frac{4\beta_{i}}{C_{v}(\beta_{i}-1)}+\frac{4\alpha_{i}}{C_{v}(1-\alpha_{i})}$$
where $C_{v}$ is a positive real number. Equation (15) is independent of the initial conditions. The observer was subsequently applied for disturbance estimation in this paper.

#### 2.2.3. Controller Design

In this section, the robust controller is designed, considering actuator saturation, faults, and external disturbances, based on the modified preassigned finite-time performance function. An auxiliary variable is introduced, as described below:(16)$$s=w_{e}+\lambda \cdot tanh(kq_{ev})$$
where λ and k are positive constants, and $s\in R^{3}$ is a sliding-mode-like vector designed to restrict the angular velocity.

**Theorem** **1.***If *$s_{i}=0$*, the attitude tracking errors *$q_{evi}$* and *$w_{ei}$* will converge to a small region* $\Omega_{1}$* around the origin within finite time *$T_{r}$.

**Proof.** If s=0, we construct the following Lyapunov function:(17)$$V_{1}=\frac{1}{2}\left[q_{ev}^{T}q_{ev}+{\left(1-q_{e0}\right)}^{2}\right]$$Consider the Lyapunov function $V_{1}$. We have ${\left(1-q_{e0}\right)}^{2}\leq \left(1-q_{e0}\right)\left(1+q_{e0}\right)=q_{ev}^{T}q_{ev}$, which shows that $V_{1}\leq q_{ev}^{T}q_{ev}$.By substituting Equation (4) into the time derivation of $V_{1}$, we obtain the following:(18)$$\begin{array}{l} {\dot{V}}_{1}=\frac{1}{2}q_{ev}^{T}w_{e}\\ \;\;\;\;\;=-\frac{1}{2}q_{ev}^{T}\lambda \cdot \tanh\left(kq_{ev}\right)\\ \;\;\;\;\;=-\frac{1}{2}\lambda \sum_{i=1}^{3}q_{evi}tanh({kq}_{evi}) \end{array}$$According to Lemmas 1 and 3, we have the following:(19)$$\begin{array}{l} {\dot{V}}_{1}\leq -\frac{1}{2}\lambda \sum_{i=1}^{3}\left(\left|q_{evi}\right|-\frac{\delta_{0}}{k}\right)\\ \;\;\;\;\;=-\frac{1}{2}\lambda \sum_{i=1}^{3}\left|q_{evi}\right|+\frac{3\lambda \delta_{0}}{2k}\\ \;\;\;\;\;\leq -\frac{1}{2}\lambda {\left(q_{ev}^{T}q_{ev}\right)}^{\frac{1}{2}}+\frac{3\lambda \delta_{0}}{2k}\\ \;\;\;\;\;\leq -\frac{1}{2}\lambda {V_{1}}^{\frac{1}{2}}+\frac{3\lambda \delta_{0}}{2k} \end{array}$$From Equation (19), it follows that ${\dot{V}}_{1}$ is negative outside of a residual set $\Omega_{1}$, where$$\Omega_{1}=\left\{\left|q_{evi}\right|\in R,\left|w_{ei}\right|\in R|\left|q_{evi}\right|<\frac{\delta_{0}}{k},\left|w_{ei}\right|<\lambda tanh(\delta_{0})\right\}$$According to Definition 3 and Equation (19), we conclude that the attitude error $q_{evi}$ will converge to a small region $\Omega_{1}$ within finite time $T_{r1}$, with $T_{r1}$ satisfying $T_{r1}\leq \frac{2\sqrt{V(q_{ev0})}}{\left(0.5\lambda -\theta \right)}$, 0<θ<0.5λ. The proof of Theorem 1 is therefore completed.If $\left\|s\right\|<\mu$, according to Equation (16), we have $\left\|w_{e}\right\|-\lambda \leq \left\|s\right\|<\mu$, and $\left\|w_{e}\right\|<\mu +\lambda$. Additionally, since $\left\|w_{e}\right\|>\left\|w\right\|-\left|R_{e}\right|\left\|w_{d}\right\|$, it follows that(20)$$\left\|w\right\|<\mu +\lambda +\left\|w_{d}\right\|$$Thus, the angular velocity limitation is satisfied if appropriate values for μ and λ are selected. □

**Remark** **1.***A sliding-mode-like vector combined with a performance function is used to avoid excessive angular velocity. The larger the parameters μ and λ, the larger the angular velocity. Spacecraft in engineering are fitted with flexible attachments, and an excessive angular velocity will cause vibration. The appropriate parameters can be selected to solve this issue in engineering applications*.

Next, the controller is designed to restrict sliding-mode-like variable *s* based on the MPFTF. The parameter $\psi_{i}$ in Equation (12) is defined as follows:(21)$$\psi_{i}=\left\{\begin{array}{c} m_{0}tanh\left(s_{i}\left(t\right)-n\rho_{i}\left(t\right)\right)\;\;\;\;\;\;\;\;\;\;\;\;\;\;\;\;\;\;\;\;\;\;\;\;\;\;\;\;\;\;\;s_{i}\left(t\right)>n\rho_{i}\left(t\right)\\ 0\;\;\;\;\;\;\;\;\;\;\;\;\;\;\;\;\;\;\;\;\;\;\;\;\;\;\;\;\;\;\;\;\;\;\;\;\;\;\;\;\;\;\;\;\;\;\;\;\;\;\;\;-n\rho_{i}\left(t\right)<s_{i}\left(t\right)<n\rho_{i}\left(t\right)\\ m_{0}tanh\left(-s_{i}\left(t\right)-n\rho_{i}\left(t\right)\right)\;\;\;\;\;\;\;\;\;\;\;\;\;\;\;\;\;\;\;\;\;\;\;\;\;s_{i}\left(t\right)<-n\rho_{i}\left(t\right) \end{array}\right.$$
where $m_{0}$ is a positive constant, and n determines the size of the safety region, n∈(0,1).

To complete the controller design, the following barrier Lyapunov function, $V_{2}$, is introduced.(22)$$V_{2}=\frac{1}{2}\sum_{i=1}^{3}\ln\frac{1}{1-e_{i}^{2}}+\frac{1}{2}z^{T}z$$

Here, $e_{i}=\frac{s_{i}}{{\bar{\rho }}_{i}}$ for *i* = 1, 2, 3. The function $V_{2}$ is valid if, and only if, $\left|e_{i}\right|<1$, which implies that $-{\bar{\rho }}_{i}<s_{i}<{\bar{\rho }}_{i}$. The time derivation of $V_{2}$ is given by(23)$${\dot{V}}_{2}=e^{T}M\dot{e}+z^{T}\dot{z}$$
where $M=diag([\frac{1}{1-e_{1}^{2}},\frac{1}{1-e_{2}^{2}},\frac{1}{1-e_{3}^{2}}])\in R^{3\times 3}$, and ${\dot{e}}_{i}=\frac{{\dot{s}}_{i}}{{\bar{\rho }}_{i}}-\frac{{\dot{\bar{\rho }}}_{i}}{{\bar{\rho }}_{i}^{2}}s_{i}$. According to Equation (16), the expression for $\dot{e}$ is(24)$$\dot{e}=diag\left(\left[\frac{1}{{\bar{\rho }}_{i}}\right]\right){\dot{w}}_{e}+\lambda diag\left(\left[\frac{1}{{\bar{\rho }}_{i}}\right]\right)\dot{\delta }-\upsilon$$
where $\dot{\delta }=kdiag\left(\left[\frac{1}{{cosh}^{2}\left(kq_{ev1}\right)},\frac{1}{{cosh}^{2}\left(kq_{ev2}\right)},\frac{1}{{cosh}^{2}\left(kq_{ev3}\right)}\right]\right){\dot{q}}_{ev}$ and(25)$$\upsilon ={\left[\frac{{\dot{\bar{\rho }}}_{1}}{{\bar{\rho }}_{1}^{2}}s_{1},\frac{{\dot{\bar{\rho }}}_{2}}{{\bar{\rho }}_{2}^{2}}s_{2},\frac{{\dot{\bar{\rho }}}_{3}}{{\bar{\rho }}_{3}^{2}}s_{3}\right]}^{T}$$

To overcome torque saturation, the variable z is introduced, as described in [45]:(26)$$\dot{z}=\left\{\begin{array}{c} -k_{z}\bar{M}e-\frac{e^{T}\bar{M}J^{-1}\Delta u}{{\left\|z\right\|}^{2}}z-\bar{k}z\;\;\;\left\|z\right\|>\sigma \\ -k_{z}\bar{M}e-\bar{k}z\;\;\;\;\;\;\;\;\;\;\;\;\;\;\;\;\;\;\;\;\;\;\;\;\;\;\;\;\;\left\|z\right\|\leq \sigma \end{array}\right.$$
where $\bar{M}=Mdiag\left(\left[\frac{1}{{\bar{\rho }}_{i}}\right]\right)$. $k_{z}\;\mathrm{and}\;\bar{k}$ are the positive gains. σ is a small positive constant. Substituting this expression into Equation (23) gives(27)$${\dot{V}}_{2}=e^{T}\bar{M}\left({\dot{w}}_{e}+\lambda \dot{\delta }-diag\left(\left[{\bar{\rho }}_{i}\right]\right)v\right)+z^{T}\dot{z}$$

Substituting Equation (5) into Equation (27) yields(28)$${\dot{V}}_{2}=e^{T}\bar{M}\left(\begin{array}{c} F+J^{-1}\left(u_{c}+∆u\right)+D_{f}+\lambda \dot{\delta }\\ -diag\left(\left[{\bar{\rho }}_{i}\right]\right)v \end{array}\right)+z^{T}\dot{z}$$

The control law $u_{c}$ is designed as(29)$$u_{c}=J(-F-{\hat{D}}_{f}-\lambda \dot{\delta }+diag\left(\left[{\bar{\rho }}_{i}\right]\right)v-k_{e}(I+{\bar{M}}^{-1})e+k_{z}z)$$
where $I\in R^{3\times 3}$ is the unit matrix. $k_{e}\in R$ is the positive gain.

#### 2.2.4. Stability Analysis

**Theorem** **2.**
*Consider the system described by Equations (3)–(5). Let the initial errors satisfy the condition $-1<e_{i}<$ 1, and assume that Assumptions 1 and 2 are satisfied. Under the control law defined in Equation (29) and the adaptive law given by Equation (26), the sliding-mode-like vector *

$s_{i}$

* will remain strictly within the preassigned performance envelope and converge to the bound *

$\rho_{Tfi}$

* within the settling time *

$T_{i}$

*. Consequently, the attitude tracking error *

$q_{evi}$

* and angular velocity tracking error *

$w_{ei}$

* will converge to a small region, *

${∆}_{q},{∆}_{w}$

*, around the origin.*


**Proof.** By substituting Equation (29) into Equation (28), and assuming that $\left\|z\right\|>\sigma$, we obtain(30)$${\dot{V}}_{2}=-e^{T}k_{e}\bar{M}e-e^{T}k_{e}e-\bar{k}z^{T}z-e^{T}\bar{M}D_{e}$$
where $D_{e}={\hat{D}}_{f}-D_{f}$ denotes the estimation error of the lumped disturbance, as observed by the disturbance observer. According to [47], the estimation error converges to a small region where $\left\|D_{e}\right\|<\gamma$ within a fixed time. Consequently, we obtain(31)$${\dot{V}}_{2}\leq -e^{T}k_{e}\bar{M}e-e^{T}k_{e}e-\bar{k}z^{T}z+\varphi_{1}$$
where $\varphi_{1}=\left\|e\right\|\left\|\bar{M}\right\|\;\gamma$. Thus, we obtain(32)$$\begin{array}{l} {\dot{V}}_{2}\leq -e^{T}k_{e}\bar{M}e-e^{T}k_{e}e-\bar{k}z^{T}z+\varphi_{1}\\ \;\;\;\;\;=-k_{e}\sum_{i=1}^{3}\left(1+\frac{1}{{\bar{\rho }}_{i}\left(1-e_{i}^{2}\right)}\right)e_{i}^{2}-\bar{k}z^{T}z+\varphi_{1} \end{array}$$If the tracking error lies within the envelope under the initial conditions, the condition $-1<e_{i}<$ 1 is satisfied. Based on Lemma 2, and assuming that $\rho_{i}<\rho_{i0}$, the following inequality holds.(33)$$\begin{array}{l} {\dot{V}}_{2}\leq -k_{e}\sum_{i=1}^{3}\frac{1}{{\bar{\rho }}_{i}\left(1-e_{i}^{2}\right)}e_{i}^{2}-\bar{k}z^{T}z+\varphi_{1}\\ \;\;\;\;\;\leq -k_{e}\sum_{i=1}^{3}\frac{1}{{\bar{\rho }}_{i}}\ln\frac{1}{1-e_{i}^{2}}-\bar{k}z^{T}z+\varphi_{1}\\ \;\;\;\;\;\leq -\frac{k_{e}}{\rho_{i0}}\sum_{i=1}^{3}\ln\frac{1}{1-e_{i}^{2}}-\bar{k}z^{T}z+\varphi_{1}\\ \;\;\;\;\;\leq -\frac{k}{2}\left(\sum_{i=1}^{3}\ln\frac{1}{1-e_{i}^{2}}+z^{T}z\right)+\varphi_{1} \end{array}$$Here, $l_{1}=2min\left(\frac{k_{e}}{\rho_{i0}},\bar{k}\right)$. Hence, we have(34)$${\dot{V}}_{2}\leq -l_{1}V_{2}+\varphi_{1}$$Thus, ${\dot{V}}_{2}<0$ when $V_{2}>\frac{\varphi_{1}}{l_{1}}$. We can conclude that the time derivative of $V_{2}$ is negative outside a compact residual set $\Omega_{2}$, defined as(35)$$\Omega_{2}=\left\{V_{2}\in R|V_{2}\leq \frac{\varphi_{1}}{l_{1}}\right\}$$Assuming that $\left\|z\right\|\leq \sigma$, the following inequality holds:(36)$$e^{T}\bar{M}J^{-1}\Delta u\leq k_{e}e^{T}e+\frac{1}{4k_{e}}{\left\|\bar{M}\right\|}^{2}{\left\|J^{-1}\right\|}^{2}{\left\|∆u\right\|}^{2}$$(37)$$z^{T}\dot{z}=-z^{T}k_{z}\bar{M}e-z^{T}\bar{k}z$$Substituting Equations (26), (29), (36) and (37) into Equation (30) obtains the following expression:(38)$${\dot{V}}_{2}\leq -k_{e}e^{T}\bar{M}e-\bar{k}z^{T}z+\varphi_{2}$$
where $\varphi_{2}=\varphi_{1}+\frac{1}{4k_{e}}{\left\|\bar{M}\right\|}^{2}{\left\|J^{-1}\right\|}^{2}{\left\|∆u\right\|}^{2}$. Assuming that the initial attitude error lies within the performance envelope, both $\left\|e\right\|$ and $\left\|\bar{M}\right\|$ are bounded. Equation (29) is nonsingular, and ∆u is also bounded. As in Equation (33), the following inequality is obtained:(39)$$\begin{array}{l} {\dot{V}}_{2}\leq -k_{e}\sum_{i=1}^{3}\frac{1}{{\bar{\rho }}_{i}\left(1-e_{i}^{2}\right)}e_{i}^{2}-\bar{k}z^{T}z+\varphi_{2}\\ \;\;\;\;\;\leq -\frac{l_{1}}{2}\left(\sum_{i=1}^{3}\ln\frac{1}{1-e_{i}^{2}}+z^{T}z\right)+\varphi_{2}\\ \;\;\;\;\;\leq -l_{1}V_{2}+\varphi_{2} \end{array}$$It is also evident that ${\dot{V}}_{2}\leq 0$ when $V_{2}>\frac{\varphi_{2}}{l_{1}}$, and the time derivative of $V_{2}$ is negative outside the compact residual set $\Omega_{3}$, which is defined as(40)$$\Omega_{3}=\left\{V_{2}\in R|V_{2}\leq \frac{\varphi_{2}}{l_{1}}\right\}$$Solving Equation (39) yields(41)$$V_{2}\left(t\right)\leq \left(V_{2}\left(0\right)-\frac{\varphi_{2}}{l_{1}}\right)e^{-l_{1}t}+\frac{\varphi_{2}}{l_{1}}$$We define the set $\Pi_{1}:=\left\{V_{2}\in R|V_{2}\leq p\right\}$ and assume that the initial condition satisfies $V_{2}(0)\leq p$ for some p>0. From Equation (39), it follows that $V_{2}(t)\leq p$ for all t>0 if $l_{1}>\frac{\varphi_{2}}{p}$. The set $\Pi_{1}$ is compact. Consequently, the closed-loop system is semi-globally stable, as demonstrated within the compact set $\Pi_{1}$. Given the boundedness of $V_{2}(t)$, it can be inferred that both z and $\ln\frac{1}{1-e_{i}^{2}}$ remain bounded. The transformed error $e_{i}(t)$ remains within the set $\Pi_{2}:=\left\{e_{i}\in R^{3}|\left|e_{i}\right|<1,i=1,\;2,\;3\right\}$ for all time.Owing to the convergence of the preassigned finite-time function, it can be concluded that the sliding-mode-like variable $s_{i}(t)$ converges to the envelope defined by the performance function boundaries within the preset time $T_{i}$. Additionally, the adaptive parameter z is uniformly ultimately bounded.The above analysis leads to the conclusion that performance can be guaranteed, provided that $\left|e_{i}\right|<1$ when the initial error lies within the performance envelope. The sliding-mode-like variable $s_{i}$ evolves strictly within the MPFPF envelope, and it follows that $\underset{t\to T_{i}}{\lim}\left|s_{i}\right|<\rho_{Tfi}$. Consequently, both $q_{evi}$ and $w_{ei}$ also converge to a small set. The proof of this claim is presented in the subsequent sections.Based on Equation (16), the following holds:(42)$$w_{ei}+\lambda \cdot \tanh\left(kq_{evi}\right)=\vartheta_{i},\;\left|\vartheta_{i}\right|<\rho_{Tfi}$$Equation (42) is rewritten in the following form:(43)$$w_{ei}+\left(\lambda -\frac{\vartheta_{i}}{\tanh\left(kq_{evi}\right)}\right)\tanh\left(kq_{evi}\right)=0$$**Case** **1.***If*$\;\lambda -\frac{\vartheta_{i}}{\tanh\left(kq_{evi}\right)}\geq 0$*, the tracking error *$q_{evi}$* is convergent. If the Lyapunov function is selected in the same form as in Equation (17), a conclusion similar to Theorem 1 can be drawn. Therefore, the tracking error *$q_{evi}\;$* is finite-time convergent with (*$T_{r2}+T_{i}$*), where the detailed form of *$T_{r2}$* is similar to that of *$T_{r1}$*. The convergence domain of *$q_{evi}$* is expressed as *$\left|q_{evi}\right|<\frac{\delta_{0}}{k}$* according to Theorem 1. Additionally, the angular velocity tracking error *$w_{ei}$* converges into a region expressed by *$\left|w_{ei}\right|<\rho_{Tfi}+\lambda \tanh\left(\delta_{0}\right)$*, according to Equation (42)*.**Case** **2.**
*If*

$\;\lambda -\frac{\vartheta_{i}}{\tanh\left(kq_{evi}\right)}<0$

*, it follows that *

$\lambda <\frac{\left|\vartheta_{i}\right|}{\left|\tanh\left(kq_{evi}\right)\right|}$

*. Consequently, the attitude error *

$q_{evi}$

* converges to the region defined by *

(44)
$$\left|q_{evi}\right|<\frac{1}{k}{tanh}^{-1}(\frac{\rho_{Tfi}}{\lambda })$$

Based on Equation (42), it can be derived that(45)$$\begin{array}{l} \left|w_{ei}\right|=\left|\vartheta_{i}-\lambda \cdot \tanh\left(kq_{evi}\right)\right|\\ \;\;\;\;\;\;\;\leq \left|\vartheta_{i}\right|+\lambda \cdot \left|\tanh\left(kq_{evi}\right)\right|\\ \;\;\;\;\;\;\;\leq \left|\vartheta_{i}\right|+\rho_{Tfi}\\ \;\;\;\;\;\;\;<2\rho_{Tfi} \end{array}$$In conclusion, according to Case 1 and Case 2, the attitude tracking errors $q_{evi}$ and $w_{ei}$ converge within finite time $T_{f}$ to the region defined by$${∆}_{q}=max\left\{\frac{\delta_{0}}{k},\frac{1}{k}{tanh}^{-1}\left(\frac{\rho_{Tfi}}{\lambda }\right)\right\},\;\;{∆}_{w}=max\left\{\rho_{Tfi}+\lambda \tanh\left(\delta_{0}\right),2\rho_{Tfi}\right\}$$ and the settling time is given as $T_{f}\leq T_{r2}+T_{i}$. The proof of Theorem 2 is thus completed. □

**Remark** **2.**
*In this paper, the convergence time of the system is divided into two parts: the reaching time $T_{i}$ and the convergence time on the sliding surface $T_{r}$. $T_{i}$ can be preset through the performance function. The convergence of the attitude errors, $q_{evi}$ and $w_{ei}$, on the sliding surface is finite-time stable, as established in Theorem 1.*


**Remark** **3.**
*Due to the presence of external and internal disturbances and actuator failures, it is nearly impossible to converge exactly to the origin. However, the attitude tracking errors $q_{evi}$ and $w_{evi}$ will converge into a small region around the origin in finite time. Based on the above analysis, the larger the parameter k, the smaller the convergence region ${∆}_{q}$, ${∆}_{w}$.*


## 3. Simulation Results and Discussion

### 3.1. Simulation Scenarios and Parameter Settings

To evaluate the effectiveness of the proposed control scheme, two scenarios are considered. The first scenario involves performing a large-angle attitude maneuver, considering angular velocity and torque limitations. In the second scenario, the controller is required to maintain precise attitude tracking despite external disturbances and actuator faults, providing insight into its robustness and adaptability to unpredictable conditions. This scenario confirms that the controller can perform reliably under real-world dynamic conditions.

The parameters are set as follows. Consider the inertia matrix of the rigid spacecraft:$$J=\left[\begin{array}{ccc} 13.2 & 0.5 & 0.6\\ 0.5 & 12.6 & 0.7\\ 0.6 & 0.7 & 13.1 \end{array}\right]kg\cdot m^{2}$$

The external disturbance is given by$$d\left(t\right)=10^{-3}\times \left[\begin{array}{c} 1+5\cos(0.1t)\\ 2+5\cos(0.2t)\\ -1+5\sin(0.1t) \end{array}\right]Nm$$

In this paper, the primary parameters are set as follows. The parameters of the preassigned finite-time function are set as $\rho_{0}=0.2$, $\rho_{tf}=0.001$, $\rho_{\lambda }=2$, $m_{0}=1$, $\alpha_{1}=\alpha_{2}=10$, and n=0.5. The parameter of the sliding-mode-like vector is set as k=5. The controller-related parameters are set as $k_{e}=0.2$, $k_{z}=2$, $\bar{k}=0.1$, and σ=0.01. The maximum torque of the actuator is 5 Nm. The parameters of the fixed-time disturbance observer are $k_{1}=k_{2}=1$, α=0.9, β=1.1, and ε=0.02.

Numerical experiments were conducted on a laptop equipped with an Intel i5-10210U 2.11 GHz CPU (TSMC, Taiwan, China) and 16 GB of RAM, using Matlab 2020b and the Ode4 solver. The simulation step was set to 0.01 s.

We compare the performance of the PFTC in [35] with the methods in this paper. The main parameters of PFTC are the same as those in [35] and given by $K_{1}=diag(0.04,\;0.04,\;0.04)$, $K_{2}=diag(0.45,\;0.45,\;0.45)$, and γ=0.01. Modified Rodrigues Parameters (MRPs) were used to denote the attitude in [35]. Euler angles are obtained with a ‘zxy’ rotation sequence, including Roll, Pitch, and Yaw.

### 3.2. Attitude Maneuver

The unit quaternions for the initial and desired states are $\left[0,\;0,\;1,\;0\right]$ and $\left[1,\;0,\;0,\;0\right]$, respectively. The initial angular velocity is $w_{0}=10^{-4}\times \left[1,1,1\right]\;deg/s$, and the desired angular velocity is $w_{d}=\left[0,0,0\right]\;deg/s$. According to (9), the upper bound of the preassigned time is $T_{f}=40\;s$, and according to (16), the controller parameter λ is set to 0.2 and 0.3.

The figures below illustrate the controller’s performance during a large-angle attitude maneuver under constraints on the control torque and angular velocity.

Figure 2 presents the results for λ=0.2. The Euler angle errors are shown in Figure 2a, and the angular velocity of the spacecraft is depicted in Figure 2b. The Euler angle error is $1\times 10^{-6}\;deg$, and the angular velocity error is $\;2\times 10^{-7}\;deg/s$ within 38 s. The maximum angular velocity reaches 11.56 deg/s. The physical limitations of the actuator are not violated, with the control torque shown in Figure 2c. The sliding-mode-like vector’s performance is shown in Figure 2d, converging to the steady-state boundary within 25 s.

Similarly, the results for λ=0.3 are shown in Figure 3. The maximum angular velocity reaches 17.26 deg/s, as illustrated in Figure 3b. The Euler angle error and angular velocity error are $1\times 10^{-6}\;deg$ and $2\times 10^{-7}\;deg/s$, respectively, within 35 s, as shown in Figure 3a,b. The sliding-mode-like vector reaches the steady boundary within 20 s, in Figure 3d. Clearly, increasing the parameter λ can shorten the convergence time; however, it also leads to a larger angular velocity. Additionally, the disturbance observer error is $1\times 10^{-3}\;Nm$, as shown in both Figure 2e and Figure 3e.

Based on the results, several conclusions can be drawn. First, the proposed controller effectively limits the angular velocity during a large-angle maneuver by selecting an appropriate parameter λ. As the parameter λ increases, the maximum angular velocity also increases. According to (20), the maximum angular velocity is limited to λ + μ, where μ represents the maximum value of the performance function. In Figure 2 and Figure 3, the maximum angular velocity is less than 0.4 rad (λ=0.2, μ=0.2) and 0.8 rad (μ=0.5, λ=0.3), respectively. The design incorporates an actuator saturation model, with a maximum torque of 5 Nm under the MPFTC scheme.

Larger initial errors result in excessive torque and angular velocity during the maneuver, presenting challenges in practical systems, such as reaction wheel and gyroscope limitations. The controller designed in this study effectively alleviates these issues to some extent.

The results using the PFTF are shown in Figure 4. It is obvious that the maximum angular velocity is 50.56 deg/s during the large-angle attitude maneuver in Figure 4b. The final Euler angle error and angular velocity error are $1\times 10^{-4}\;deg$ and $2\times 10^{-6}\;deg/s$ after 38 s. Due to the excessive angular velocity, attitude stabilization becomes difficult.

### 3.3. Attitude Tracking

#### 3.3.1. Attitude Tracking with Sudden External Disturbance

In this section, an example of attitude tracking with a sudden external disturbance is presented to validate the robustness of MPFTC. The initial unit quaternion is $q_{0}=\left[1,\;0,\;0,\;0\right]$, and the initial angular velocity is $w_{0}=\left[0,\;0,\;0\right]$ rad/s. The initial desired unit quaternion is $q_{d0}=\left[1,\;0,\;0,\;0\right]$, and the desired angular velocity is $w_{d}=\left[0.03\cos\left(t/40\right),\;-0.03\sin\left(t/50\right),\;-0.03\cos\left(t/60\right)\right]\;rad/s$. To assess the effectiveness of MPFTC, a sudden disturbance is introduced, defined as $d_{a}=\left[0.9,\;0.9,\;-0.9\right]\;Nm$ for 15 s≤t≤16 s. The preset time is $T_{f}=10\;s$, with all other controller parameters remaining unchanged.

The simulation results under the PFTF and MPFTF are presented above. The results of the MPFTF are illustrated in Figure 5. In contrast, the PFTF is defined in [35], with the corresponding results shown in Figure 6.

The results indicate that the attitude tracking error exceeds the predefined steady-state boundary due to a sudden external disturbance at 15 s, as illustrated in Figure 6a. Furthermore, control singularity occurs, leading to divergence in attitude tracking, as illustrated in Figure 6b,c after 15 s.

In response, MPFTC relaxes the performance boundary, ensuring that the boundary constraints are satisfied and preventing control singularities, as shown in Figure 5d. Consequently, the control torque does not reach saturation, as illustrated in Figure 5c. As depicted in Figure 5a,b, the final angular velocity tracking error and Euler angle tracking error are $5\times 10^{-4}\;deg/s$ and $1\times 10^{-3}\;deg$, respectively, at the steady state. These results confirm that the performance of the closed-loop system is maintained even in the presence of a sudden external disturbance.

In contrast, due to the rigid boundary, the constrained variable, represented by MRPs in [35] or s in this paper, approaches the predetermined boundary, resulting in the controller outputting a large control torque. Due to the saturation of the actuator, the control command cannot be executed, leading to control divergence. The constrained variable fails to remain within the predefined envelope. The MPFTF can adjust the boundary, keeping the variables within it, as shown in Figure 5d.

#### 3.3.2. Attitude Tracking with Actuator Faults

This section presents an example involving actuator faults, with all other controller parameters remaining unchanged. No sudden external disturbance is applied, but actuator faults are considered. The parameters for the actuator faults are $e_{1}=e_{2}=e_{3}=\left\{\begin{array}{c} 1\;\;\;\;\;\;\;\;\;\;\;\;t<15\;s\\ 0.85\;\;\;\;\;\;\;t\geq 15\;s \end{array}\right.$, ${\bar{u}}_{1}=\left\{\begin{array}{c} 0\;\;\;\;\;\;\;\;\;\;\;\;\;t<15\;s\\ 0.9\;\;\;\;\;\;\;\;\;\;\;t\geq 15\;s \end{array}\right.$, ${\bar{u}}_{2}=\left\{\begin{array}{c} 0\;\;\;\;\;\;\;\;\;\;\;\;\;t<15\;s\\ -0.9\;\;\;\;\;\;\;\;t\geq 15\;s \end{array}\right.$, and ${\bar{u}}_{3}=\left\{\begin{array}{c} 0\;\;\;\;\;\;\;\;\;\;\;\;\;t<15\;s\\ 0.9\;\;\;\;\;\;\;\;\;\;\;t\geq 15\;s \end{array}\right.$. The simulation results are presented below.

In Figure 7 and Figure 8, the results of attitude tracking are presented using the MPFTF and PFTF, respectively, in the presence of actuator faults. The constraint variables are influenced by the actuator fault. However, in Figure 7a, the controller, based on the MPFTF, is able to stabilize the sliding-mode-like vector s within 1 s in the presence of actuator faults. The attitude velocity tracking error and Euler angle error remain within $5\times 10^{-4}\;deg/s$ and $1\times 10^{-3}\;deg$, respectively, as depicted in Figure 7b,c. Due to the adaptive adjustment of the boundary, the control torque, though increased, does not reach saturation, as shown in Figure 7d.

In contrast, the results based on the PFTF show that control divergence occurs due to the rigid boundary, as illustrated in Figure 8. These results are consistent with those presented in Section 3.3.1 and shown in Figure 6. The above results demonstrate that the proposed controller guarantees superior tracking performance, even in the presence of actuator faults.

## 4. Conclusions

A modified preassigned finite-time controller was designed for large-angle attitude maneuvering and tracking, considering angular velocity constraints and actuator limitations. By constraining the sliding-mode-like vector, the controller limits the angular velocity, and an estimate of the maximum angular velocity is provided. Actuator saturation is adaptively compensated. The performance boundaries are adaptively adjusted to counteract sudden external or internal disturbances, thereby avoiding control singularity. This approach is superior to the rigid boundary-based performance function. The proposed controller was shown to stabilize attitude errors within a neighborhood of the origin, achieving stabilization within a finite time horizon. The simulation results demonstrated that, during attitude maneuvering, the attitude error and angular velocity error were $1\times 10^{-6}\;deg$ and $2\times 10^{-7}\;deg/s$, respectively. Excessive angular velocity is prevented by adjusting the parameter λ, a strategy not implemented in the previous literature. During attitude tracking, the attitude error and angular velocity error are $1\times 10^{-3}\;deg$ and $5\times 10^{-4}\;deg/s$, respectively, even in the presence of sudden external and internal disturbances. Overall, the simulation results validate the effectiveness of the proposed controller.

## Figures and Tables

**Figure 1 sensors-25-00986-f001:**
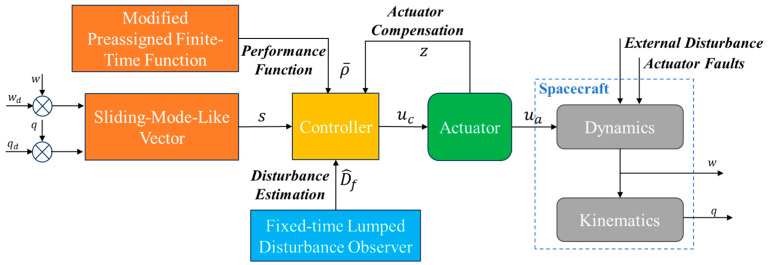
Block diagram of the closed-loop system.

**Figure 2 sensors-25-00986-f002:**
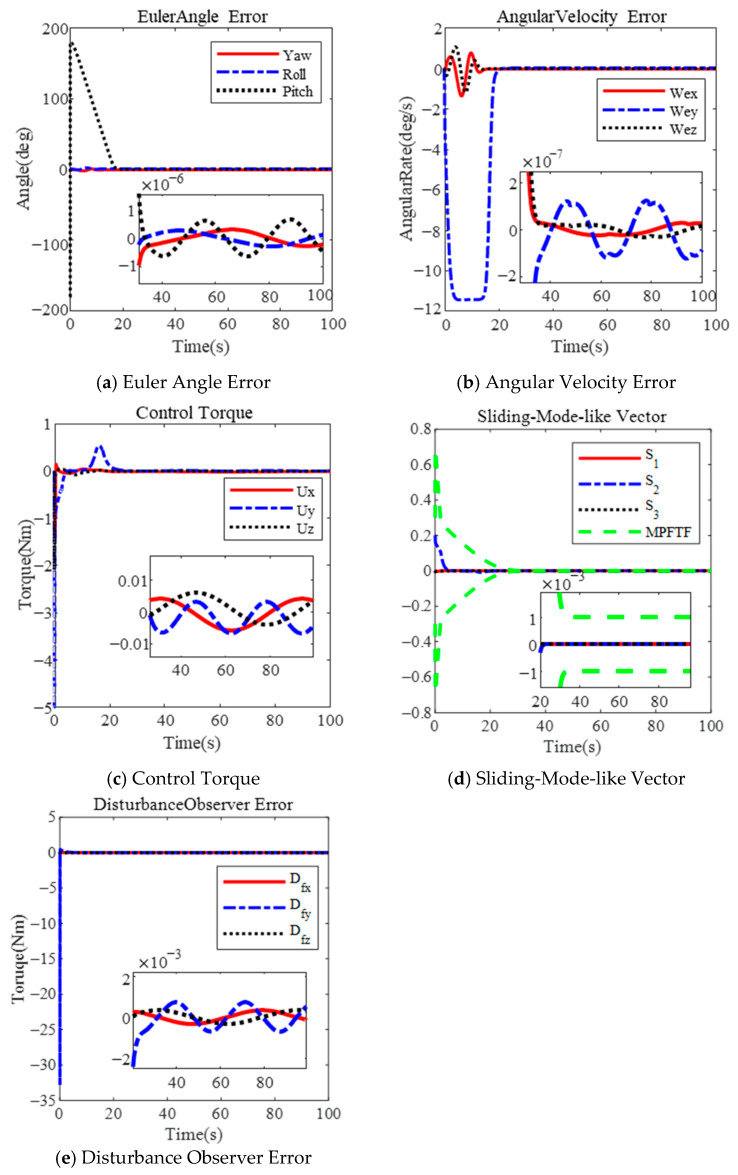
Large-angle attitude maneuver based on MPFTF for λ=0.2.

**Figure 3 sensors-25-00986-f003:**
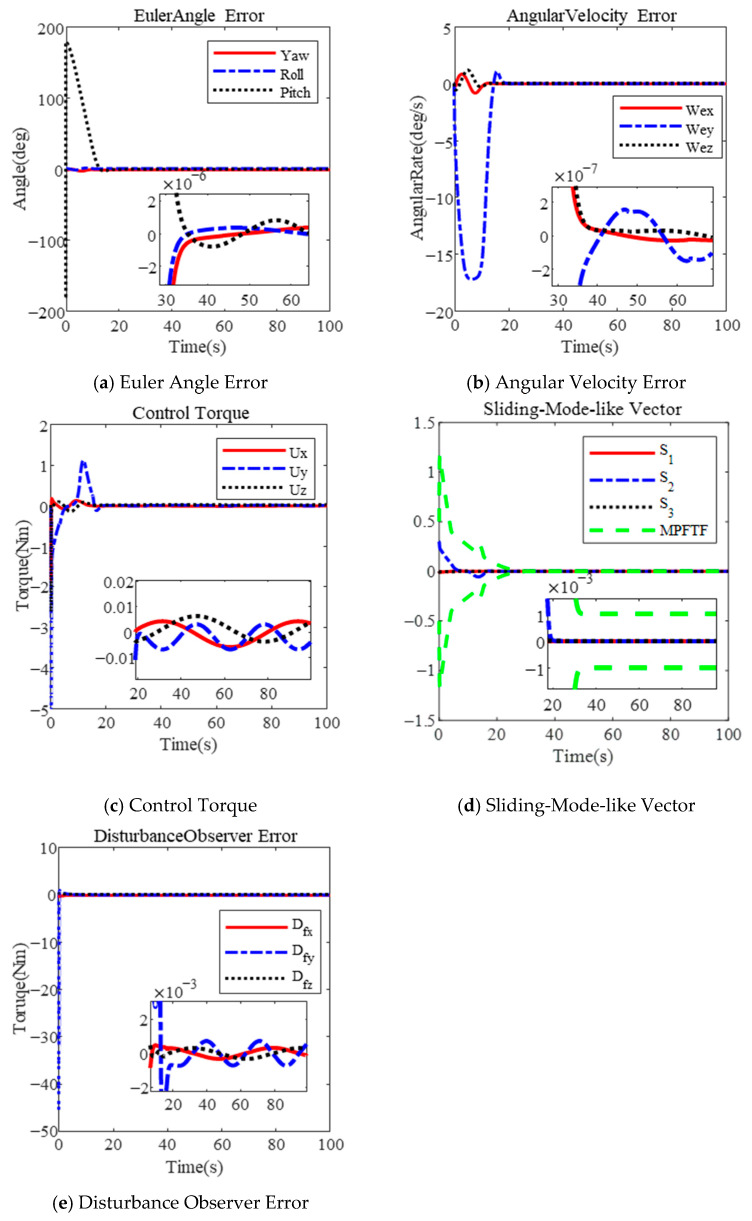
Large-angle attitude maneuver based on MPFTF for λ=0.3.

**Figure 4 sensors-25-00986-f004:**
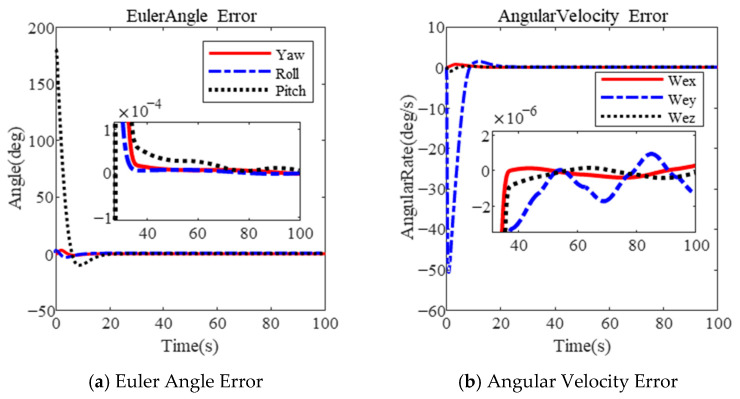
Large-angle attitude maneuver based on PFTF in [35].

**Figure 5 sensors-25-00986-f005:**
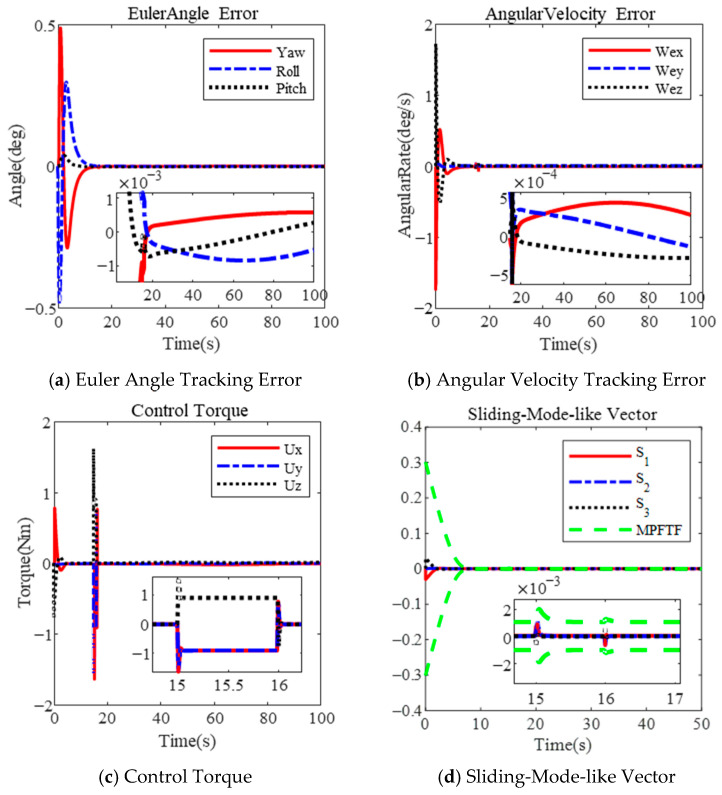
Attitude tracking with sudden external disturbance based on MPFTF.

**Figure 6 sensors-25-00986-f006:**
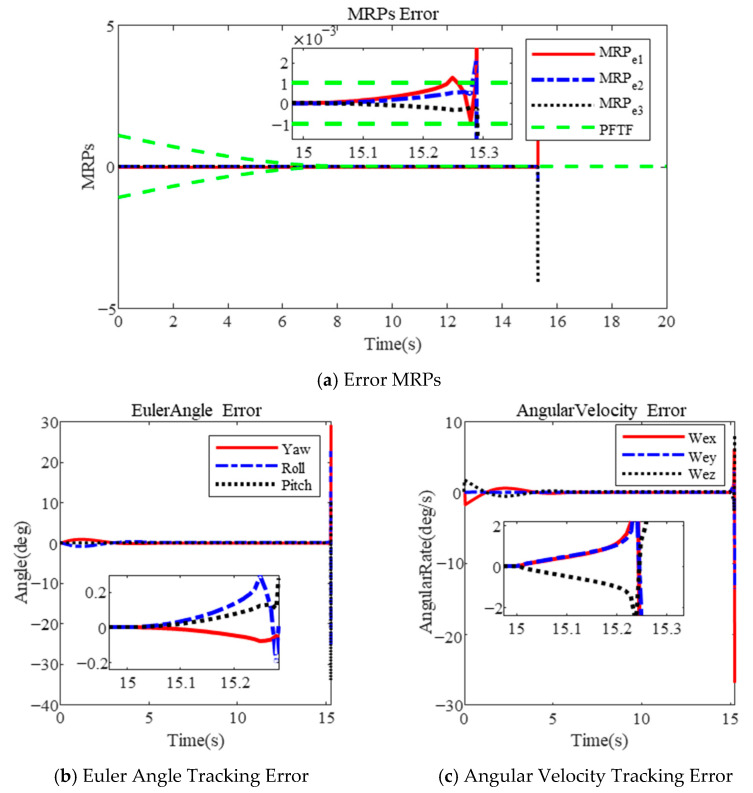
Attitude tracking with sudden external disturbance based on PFTF in [35].

**Figure 7 sensors-25-00986-f007:**
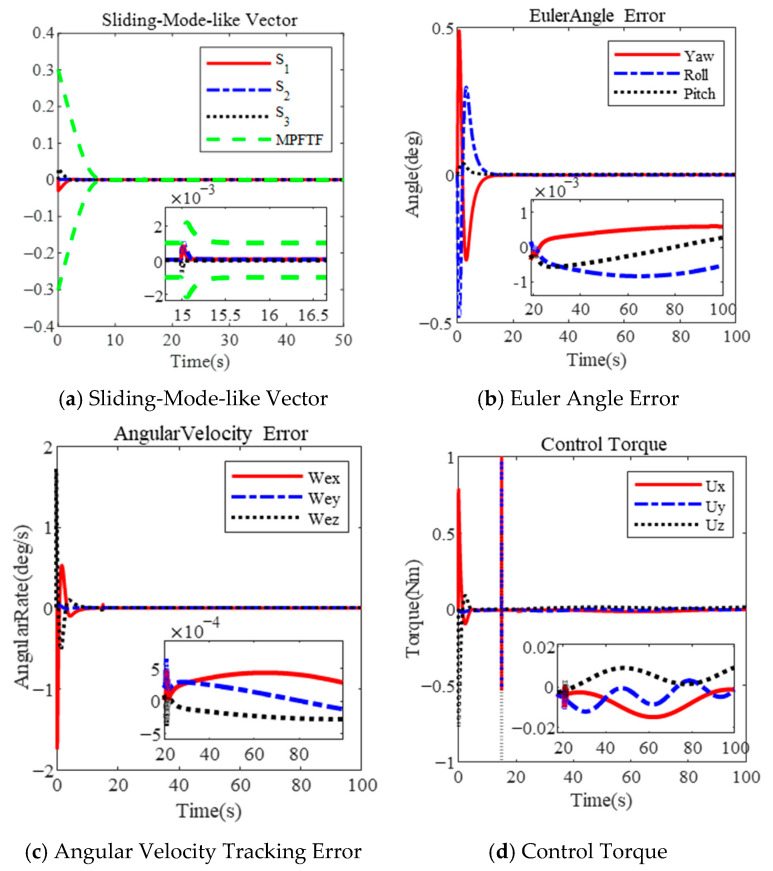
Attitude tracking with actuator faults based on MPFTF.

**Figure 8 sensors-25-00986-f008:**
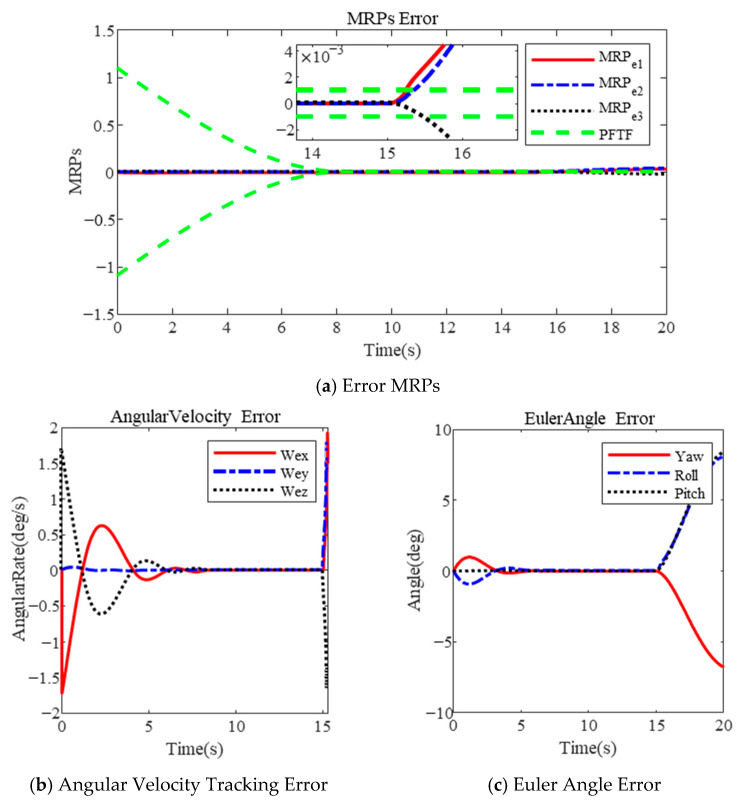
Attitude tracking with actuator faults based on PFTF in [35].

## Data Availability

The original contributions presented in this study are included in this article; further inquiries can be directed to the corresponding author.

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
