# Peer review of "A Modified Preassigned Finite-Time Control Scheme for Spacecraft Large-Angle Attitude Maneuvering and Tracking"

_sensors, 2025, doi:10.3390/s25030986_

Round 1
Reviewer 1 Report
Comments and Suggestions for Authors
1 There is already a large literature on prescribed performance control (PPC) methods. The PPC method proposed in this paper that considers actuator saturation has also been heavily studied. The paper should default to the premise that the system is input-output stabilized. The real question worth investigating is can the steady state value of the performance envelope be arbitrarily small?
2 Does the introduction section need to answer what is the essential difference between PPC, barrier Lyapunov functions and funnel control?
3 Control algorithm simulation results are beneficial, what are the risk points on engineering landing?
Reviewer 2 Report
Comments and Suggestions for Authors
This paper discusses large-angle attitude maneuver and tracking for rigid spacecraft, addressing various challenges such as angular velocity and torque constraints, actuator faults, and external disturbances. However, the paper still has some deficiencies:
- Please highlight the innovations and key points of the paper in the abstract, and emphasize the advantages of the proposed method in engineering applications.
- The authors should carefully review the paper for logic and language to ensure that the core ideas are accurately expressed.
- In the "Results and Discussion" section, regarding the determination of parameters such as λ, the authors should provide further explanation on the selection of these parameters.
- It is recommended to add a subsection in Section 3 specifically introducing the simulation data and related parameters.
- I believe I have not found any discussion regarding the preset time horizon in the paper. I am particularly concerned about whether the efficiency in the simulation experiments can meet the requirements of practical applications. I hope the revised manuscript can provide further clarification on this point.
Reviewer 3 Report
Comments and Suggestions for Authors
This paper develops a finite-time attitude controller for rigid spacecraft maneuvers with large angles, addressing constraints on angular velocity and torque, actuator faults, and external disturbances. It introduces a sliding-mode-like vector and a modified finite-time function to regulate control. Utilizing a barrier Lyapunov function, the controller stabilizes the closed-loop system, adaptively compensates for actuator saturation, and employs a fixed-time disturbance observer to estimate disturbances. The closed-loop system's finite-time stability is demonstrated, and simulations confirm the controller's effectiveness.
The introduction is effective. Please consider including "Design and Analysis of a Growable Artificial Gravity Space Habitat," "Review of Space Habitat Designs for Long-Term Space Explorations," and "In-Space Fabrication and Growth of Affordable Large Interior Rotating Habitats" to provide a broader scope and enhance applicability to spacecraft applications. References 23 and 24 are cited frequently; please clearly articulate this paper's unique contribution in the conclusion.
References are missing for Equations (1) and (2).
Equations should be accompanied by more detailed explanations.
The variables in Figure 1 should be defined clearly.
Lines 166-175 require additional details.
Some details of the experimental setup are missing.
Please add a picture of the experimental setup to the numerical example.
The conclusion is suggested to include statements supported by values from your numerical experiments.
The discussion of the figures should be more detailed.
The units appear unusual; please check for and correct any typos.
Comments on the Quality of English Language- The manuscript would benefit from a thorough review of the English language.
Reviewer 4 Report
Comments and Suggestions for Authors
The manuscript " A modified preassigned finite-time control scheme for space-craft large-angle attitude maneuver and tracking" proposes a novel attitude controller based on the Barrier Lyapunov Function. The angular velocity constraint is incorporated by constructing a sliding-mode-like vector, which is subsequently constrained via the performance function. Furthermore, an adaptive term is employed to compensate for actuator saturation. A modified preassigned finite-time function, incorporating an adaptive term, is introduced to address control singularity cause by sudden external or internal disturbances in steady-state. The idea is potentially interesting, but this manuscript still has a lot of room for improvement. The main modification suggestions are as follows.
(1) The author should write carefully with reference to the template and try to avoid errors in format and English expression.
(2) The author should check the full text formula, seems unable to distinguish scalar and vector.
(3) The author should analyze the simulation results in detail, so that readers can clearly understand the superiority of the proposed method.
(4) Comparative simulations with traditional methods should be added.
Round 2
Reviewer 2 Report
Comments and Suggestions for Authors
I have no additional questions.
Reviewer 3 Report
Comments and Suggestions for Authors
The authors have satisfactorily addressed all of my concerns. I recommend that the paper be accepted in its current form.
Reviewer 4 Report
Comments and Suggestions for Authors
The author has made detailed modifications to the recommendations, and the current version can be accepted for publication.